# A Human-Following Motion Planning and Control Scheme for Collaborative Robots Based on Human Motion Prediction

**DOI:** 10.3390/s21248229

**Published:** 2021-12-09

**Authors:** Fahad Iqbal Khawaja, Akira Kanazawa, Jun Kinugawa, Kazuhiro Kosuge

**Affiliations:** 1Center for Transformative AI and Robotics, Graduate School of Engineering, Tohoku University, Sendai 980-8579, Japan; kanazawa@irs.mech.tohoku.ac.jp (A.K.); kinugawa@irs.mech.tohoku.ac.jp (J.K.); or kosuge@hku.hk (K.K.); 2Robotics and Intelligent Systems Engineering (RISE) Laboratory, Department of Robotics and Artificial Intelligence, School of Mechanical and Manufacturing Engineering (SMME), National University of Sciences and Technology (NUST), Sector H-12, Islamabad 44000, Pakistan; 3Department of Electrical and Electronic Engineering, The University of Hong Kong, Pokfulam, Hong Kong

**Keywords:** human–robot interaction, human–robot collaboration, collaborative robots, motion planning, robot control, human motion prediction, human-following robots

## Abstract

Human–Robot Interaction (HRI) for collaborative robots has become an active research topic recently. Collaborative robots assist human workers in their tasks and improve their efficiency. However, the worker should also feel safe and comfortable while interacting with the robot. In this paper, we propose a human-following motion planning and control scheme for a collaborative robot which supplies the necessary parts and tools to a worker in an assembly process in a factory. In our proposed scheme, a 3-D sensing system is employed to measure the skeletal data of the worker. At each sampling time of the sensing system, an optimal delivery position is estimated using the real-time worker data. At the same time, the future positions of the worker are predicted as probabilistic distributions. A Model Predictive Control (MPC)-based trajectory planner is used to calculate a robot trajectory that supplies the required parts and tools to the worker and follows the predicted future positions of the worker. We have installed our proposed scheme in a collaborative robot system with a 2-DOF planar manipulator. Experimental results show that the proposed scheme enables the robot to provide anytime assistance to a worker who is moving around in the workspace while ensuring the safety and comfort of the worker.

## 1. Introduction

The concept of collaborative robots was introduced in the early 1990s. The first collaborative system was proposed by Troccaz et al. in 1993 [1]. This system uses a passive robot arm to ensure safe operation during medical procedures. In 1996, Colgate et al. developed a passive collaborative robot system and applied it to the vehicle’s door assembly process carried out by a human worker [2]. In 1999, Yamada et al. proposed a skill-assist system to help a human worker carry a heavy load [3].

The collaborative robot systems are being actively introduced in the manufacturing industry. The International Organization of Standardization (ISO) amended its robot safety standards ISO 10128-1 [4] and 10128-2 [5] in 2011 to include the safety guidelines for human-robot collaboration. This led to an exponential rise in collaborative robot research and development. Today, many companies are manufacturing their own versions of collaborative robots, and these robots are being used in industries all over the world. Collaborative robots are expected to play a major role in the Industry 5.0 environments where people will work together with robots and smart machines [6].

In 2010, a 2-DOF co-worker robot “PaDY” (in-time Parts and tools Delivery to You robot) was developed in our lab to assist a factory worker in an automobile assembly process [7]. This process comprises a set of assembly tasks that are carried out by a worker while moving around the car body. PaDY assists the worker by delivering the necessary parts and tools to him/her for each task. The original control system of PaDY was developed based on a statistical analysis of the worker’s movements [7].

Many studies have been carried out on the human–robot collaborative system. Hawkins et al. proposed an inference mechanism of human action based on a probabilistic model to achieve wait-sensitive robot motion planning in 2013 [8]. D’Arpino et al. proposed fast target prediction of human reaching motion for human–robot collaborative tasks in 2015 [9]. Unhelkar et al. designed a human-aware robotic system, in which human motion prediction is used to achieve a safe and efficient part-delivery task between the robot and the stationary human worker in 2018 [10]. A recent survey on the sensors and techniques used for human detection and action recognition in industrial environments can be seen in [11]. The studies cited above [8,9,10] have improved efficiency of the collaborative tasks by incorporating human motion prediction into robot motion planning.

Human motion prediction was also introduced to PaDY. In 2012, the delivery operation delay of the robot–human handover tasks was reduced by utilizing prediction of the worker’s arrival time at the predetermined working position [12]. In 2019, a motion planning system was developed which optimized a robot trajectory by taking the prediction uncertainty of the worker’s movement into account [13]. In those studies [12,13], the robot repeats the delivery motion from its home position to each predetermined assembly position. If the robot can follow the worker during the assembly process, the worker can pick up necessary parts and tools from the robot at any time. Thus, more efficient collaborative work could be expected by introducing human-following motion.

In this paper, human-following motion of the collaborative robot is proposed for delivery of parts and tools to the worker. The human-following collaborative robot system needs to stay close to the human worker, while avoiding collision with the worker under the velocity and acceleration constraints. The contribution of this paper is summarized as follows:1The proposed human-following motion planning and control scheme enables the worker to pick up the necessary parts and tools when needed.2The proposed scheme achieves the human-following motion with a sufficiently small tracking error without adversely affecting the safety and comfort of the worker.3Experiments conducted in an environment similar to a real automobile assembly process illustrate the effectiveness of the proposed scheme.

This is a quantitative study where we conducted the experiments ourselves and analyzed the data collected from these experiments to deduce the results. The proposed scheme predicts the motion of the worker and calculates an optimal delivery position for the handover of parts and tools from the worker to the robot for each task of the assembly process. This scheme has been designed for a single worker operating within his/her workspace. It is not designed for the cases when multiple workers are operating in the same workspace, or when the worker moves beyond the workspace.

The rest of the paper is organized as follows. Section 2 describes the related works. Section 3 gives an overview of the proposed scheme, including the delivery position determination, the worker’s motion prediction, and trajectory planning and control scheme. The experimental results are discussed in Section 4. Section 5 concludes this paper.

## 2. Related Works

In this section, we present a review of the existing research on human–robot handover tasks, human-following robots, and motion/task planning based on human motion prediction.

### 2.1. Human–Robot Handover

Some studies have considered the problem of psychological comfort of the human receiver during the handover task. Baraglia et al. addressed the issue of whether and when a robot should take the initiative [14]. Cakmak et al. advocated the inclusion of user preferences while calculating handover position [15]. They also identified that a major cause of delay in the handover action is the failure to convey the intention and timing of handing over the object from the robot to the human [16]. Although these studies deal with important issues for improving the human–robot collaboration, it is still difficult to apply them in actual applications because psychological factors cannot be directly observed.

Some other studies used observable physical characteristics of the human worker for planning a robot motion that is safe and comfortable for the worker. Mainprice et al. proposed a motion planning scheme for human–robot collaboration considering HRI constraints such as constraints of distance, visibility and arm comfort of the worker [17]. Aleotti et al. devised a scheme in which the object is delivered in such a way that its most easily graspable part is directed towards the worker [18]. Sisbot et al. proposed a human-aware motion planner that is safe, comfortable and socially acceptable for the human worker [19].

The techniques and algorithms mentioned above operate with the assumption that the worker remains stationary in the environment. To solve the problem of providing assistance to a worker who moves around in the environment, we propose a human-following approach with HRI constraints in this paper.

### 2.2. Human-Following Robots

Several techniques have been proposed to carry out human-following motion in various robot applications. One of the first human-following approaches was proposed by Nagumo et al., in which an LED device carried by the human was detected and tracked by the robot using a camera [20].

Hirai et al. performed visual tracking of the human back and shoulder in order to follow a person [21]. Yoshimi et al. used several parameters including the distance, speed, color and texture of human clothes to achieve stable tracking in complex situations [22]. Morioka et al. used the reference velocities for human-following control calculated from estimated human position under the uncertainty of the recognition [23]. Suda et al. proposed a human–robot cooperative handling control using force and moment information [24].

The techniques cited in this section focus on performing human-following motion of the robot to achieve safe and continuous tracking. However, these schemes use the feedback of the observed/estimated current position of the worker. This makes it difficult for the robot to keep up with the worker who is continuously moving around in the workspace. In this paper, we solve this problem by applying human motion prediction and MPC.

### 2.3. Motion/Task Planning Based on Human Motion Prediction

In recent years, many studies have proposed motion planning using human motion prediction. The predicted human motion is used to generate a safe robot trajectory. Mainprice et al. proposed to plan a motion that avoids the predicted occupancy of the 3D human body [25]. Fridovich-Keil et al. proposed to plan a motion that avoids the risky region calculated by the confidence-aware human motion prediction [26]. Park et al. proposed a collision-free motion planner using the probabilistic collision checker [27].

Several studies have proposed robot task planning to achieve collaborative work based on human motion prediction. Maeda et al. achieved a fluid human–robot handover by estimating the phase of human motion [28]. Liu et al. presented a probabilistic model for human motion prediction for task-level human–robot collaborative assembly [29]. Cheng et al. proposed an integrated framework for human–robot collaboration in which the robot perceives and adapts to human actions [30].

Human motion prediction has been effectively used in various problems of human–robot interaction. In this paper, we apply the human motion prediction to human-following motion of the collaborative robot for delivery of parts/tools to a worker.

## 3. Proposed Motion Planning and Control Scheme

### 3.1. System Architecture

Figure 1 shows the system architecture of our proposed scheme. This scheme consists of three major parts:1Delivery position determination;2Worker’s motion prediction;3Trajectory planning and control.

In delivery position determination, an optimal delivery position is estimated using an HRI-based cost function. This cost function is calculated using the skeletal data of the worker measured by the 3-D vision sensor. In workers’ motion prediction, the position data obtained from the vision sensor are used to predict the motion of the worker. Moreover, after the completion of each work cycle, the worker’s model is updated using the stored position data. In the trajectory planning step, an optimal trajectory from the robot’s current position to the goal position is calculated using the receding horizon principle of Model Predictive Control (MPC). The robot motion controller ensures that the robot follows the calculated trajectory. The detailed description of these three parts of our scheme is given in the subsequent subsections.

### 3.2. Delivery Position Determination

In our proposed scheme, the delivery position is determined by optimizing an HRI-based cost function. This cost function includes terms related to the safety, visibility and arm comfort of the worker. These terms are calculated from the worker’s skeletal data observed by the 3-D vision sensor in real-time. This concept was first introduced by Sisbot et al. for motion planning of mobile robots [31]. The analytical form of the HRI-based cost function was proposed in our previous study [32]. Here, we provide a brief description of the cost function and solver for determining the optimal delivery position.

Let $p_{\mathrm{del}}\in R^{n}$ be the *n* dimensional delivery position, then the total cost $Cost(p_{\mathrm{del}},s_{w})$ is expressed as:$$\begin{array}{c} Cost\left(p_{\mathrm{del}},s_{w}\right)=C_{V}\left(p_{\mathrm{del}},s_{w}\right)+C_{S}\left(p_{\mathrm{del}},s_{w}\right)+C_{A}\left(p_{\mathrm{del}},s_{w}\right) \end{array}$$
where $s_{w}$ is latest sample of the worker’s skeletal data obtained from the sensor. $C_{V}\left(p_{\mathrm{del}},s_{w}\right)$ is the visibility cost that maintains the delivery position within the visual range of the worker. This cost is expressed as a function of the difference between the worker’s body orientation and the direction of the delivery position with respect to the worker’s body center. $C_{S}\left(p_{\mathrm{del}},s_{w}\right)$ is the safety cost that prevents the robot from colliding with the worker. This cost is expressed as a function of the distance between the worker’s body center and the delivery position. $C_{A}\left(p_{\mathrm{del}},s_{w}\right)$ is the arm comfort cost that maintains the delivery position within the suitable distance and orientation for the worker. This cost is a function of the joint angles of the worker’s arm. In addition, this cost penalizes the delivery position where the worker needs to use his/her non-dominant hand.

The optimal delivery position is calculated by minimizing the cost function $Cost(p_{\mathrm{del}}$$,s_{w})$. Since $Cost(p_{\mathrm{del}},s_{w})$ is a non-convex function, we use Transition-based Rapidly-exploring Random Tree(T-RRT) method [33] to find the globally optimal solution. We apply T-RRT only in the vicinity of the worker to calculate the optimal solution in real-time. The process of determining the optimal delivery position is summarized in Algorithm 1.
**Algorithm 1:** Determination of Optimal Delivery Position using T-RRT**Input:** Worker’s position $p_{w}$,
 Current sample of the worker’s skeleton $s_{w}$, Sampling range $r_{s}$, HRI cost function $Cost(p_{\mathrm{del}},s_{w})$**Output:** Optimal delivery position $p_{\mathrm{del}}$
1: Set the sampling area $S_{\mathrm{near}}$ using $p_{w}$ and $r_{s}$2: $p_{\mathrm{cur}}\leftarrow Sample\left(S_{\mathrm{near}}\right)$3: $Cost_{\mathrm{cur}}\leftarrow Cost\left(p_{\mathrm{cur}},s_{w}\right)$4: Counter←05: **while**$\;Counter\leq Counter_{\max}$
**do**6:  $p_{\mathrm{rand}}\leftarrow Sample\left(S_{\mathrm{near}}\right)$7:  $p_{\mathrm{new}}\leftarrow p_{\mathrm{cur}}+\delta \left(p_{\mathrm{rand}}-p_{\mathrm{cur}}\right)$8:  $Cost_{\mathrm{new}}\leftarrow Cost\left(p_{\mathrm{new}},s_{w}\right)$9:  **if** $TransitionTest(Cost_{\mathrm{new}},Cost_{\mathrm{cur}},d_{\mathrm{new}-\mathrm{cur}})$ **then**10:   $p_{\mathrm{cur}}\leftarrow p_{\mathrm{new}}$11:   $Cost_{\mathrm{cur}}\leftarrow Cost_{\mathrm{new}}$12:   Counter←013:  **else**14:   Counter←Counter+115:  **end if**16: **end while**17: $p_{\mathrm{del}}\leftarrow p_{\mathrm{cur}}$18: **return**$\;p_{\mathrm{del}}$

Figure 2 shows an example of the cost map in the workspace around the worker calculated from the HRI constraints. The worker’s shoulder positions (red squares) and the calculated delivery position (green circle) are shown in the figure. We can see that the proposed solver can calculate the delivery position that has the minimum cost in the cost map.

### 3.3. Worker’s Motion Prediction

The worker’s motion is predicted by using Gaussian Mixture Regression (GMR) proposed in our previous work [34]. GMR models the worker’s past movements and predicts his/her future movements in the workspace. Here, we provide a brief description of the motion prediction using GMR.

Suppose that $p_{c}=p_{w}^{\left(t\right)}\in R^{n}$ is the worker’s current position at time step *t*, $p_{h}={\left(p_{w}^{\left(t-1\right)}\;p_{w}^{\left(t-2\right)}\;\cdots p_{w}^{\left(t-d\right)}\right)}^{T}\in R^{n\times (d-1)}$ is the position history, and *d* is the length of the position history. GMR models the conditional probability density function $p_{r}\left(p_{c}|p_{h}\right)$ whose expectation $E[p_{c}|p_{h}]$ means the worker’s future position and variance $V[p_{c}|p_{h}]$ is the uncertainty of the prediction. The details of GMR calculation are shown in Appendix B.

The procedure for the long-term motion prediction using GMR is summarized in Algorithm 2. The calculation to predict the worker’s position at the next time step is repeated until the length of the predicted trajectory becomes equal to the maximum prediction length $T_{p}$. The worker’s predicted motion is expressed as the sequence of Gaussian distributions $\left(N_{w}^{\left(t_{c}\right)},N_{w}^{\left(t_{c}+1\right)},\cdots N_{w}^{\left(t_{c}+T_{p}\right)}\right)$ starting from the current time $t_{c}$. $N_{w}^{\left(t\right)}$ is the worker’s predicted position distribution at step *t* expressed as:(1)$$\begin{array}{c} N_{w}^{\left(t\right)}=N\left(\mu_{w}^{\left(t\right)},\;\Sigma_{w}^{\left(t\right)}\right) \end{array}$$
where $\mu_{w}^{\left(t\right)}$ is the mean vector and $\Sigma_{w}^{\left(t\right)}$ is the covariance matrix of worker’s predicted position at step *t*.

If the worker repeats his/her normal movement, which is indicated in the process chart of the assembly process, our prediction system can predict the worker’s movement accurately enough for the system. According to our previous research, the RMSE (Root Mean Square Error) of the worker’s movement was about 0.3 m [34]. The RMSE was calculated by the comparison between the initial predicted worker’s movement and the observed worker’s movement when the worker started to move to the next working position.

When the worker moves differently from his/her normal movement, it is not easy to ensure the accuracy of the prediction. However, the proposed system operates safely even in this case, since the variance of the predicted position of the worker is included in the cost function used in the motion planning as shown in our previous study [13].
**Algorithm 2:** Worker’s motion prediction using GMR**Input:** Current time $t_{c}$,
 Current position $p_{c}^{\left(t_{c}\right)}$, Position history $p_{h}^{\left(t_{c}\right)}$, Max prediction length $T_{p}$**Output:** Predicted trajectory $\left(N^{\left(t_{c}\right)},N^{\left(t_{c}+1\right)},\cdots ,N^{\left(t_{c}+T_{p}\right)}\right)$1: **while**$\;k=1\;to\;T_{p}$
**do**2:  $p_{h}^{\left(t_{c}+k\right)}={\left(p_{c}^{\left(t_{c}+k-1\right)}\;p_{c}^{\left(t_{c}+k-2\right)}\;\cdots p_{c}^{\left(t_{c}+k-d-1\right)}\right)}^{T}$3:  $\mu_{w}^{\left(t_{c}+k\right)}=E\left[p_{c}^{\left(t_{c}+k\right)}|p_{h}^{\left(t_{c}+k\right)}\right]$4:  $\Sigma_{w}^{\left(t_{c}+k\right)}=V\left[p_{c}^{\left(t_{c}+k\right)}|p_{h}^{\left(t_{c}+k\right)}\right]$5:  $p_{c}^{\left(t_{c}+k\right)}=E\left[p_{c}^{\left(t_{c}+k\right)}|p_{h}^{\left(t_{c}+k\right)}\right]$6: **end while**

### 3.4. Trajectory Planning and Control

Figure 3 shows the concept of human-following motion planning using the worker’s predicted motion. The sequence of the worker’s predicted position distributions $(N_{w}^{\left(t_{c}\right)},$$N_{w}^{\left(t_{c}+1\right)},\cdots N_{w}^{\left(t_{c}+T_{p}\right)})$ is given to the trajectory planner. The sequence of the robot states, that is the robot trajectory $\left(q^{\left(t_{c}\right)},q^{\left(t_{c}+1\right)},\cdots q^{\left(t_{c}+T_{o}\right)}\right)$, is calculated so that the robot’s state at each time step follows the corresponding predicted position of the worker.

To achieve the prediction-based human-following robot motion, we use an MPC-based planner to consider the evaluation function for finite time future robot states. This is a well-known strategy and is often used in real-time robot applications such as task-parametrized motion planning [35] and multi-agent motion planning [36].

The cost function used in MPC consists of terminal cost and stage cost. The terminal cost deals with the cost at the terminal state of the robot, which is the delivery position in our case. Stage cost considers the state of the robot during the whole trajectory from the current configuration to the goal configuration. A distinct feature of our scheme is that the optimal delivery position, found by optimizing the HRI-based cost function, is used to calculate the terminal cost. In addition, the predicted trajectory of the worker is used to calculate the stage cost. This scheme plans the collision-free robot trajectory that follows the moving worker efficiently under the safety cost constraint and the robot’s velocity and acceleration constraints.

The cost function *J* used for the optimization of the proposed trajectory planner is expressed as:(2)$$\begin{array}{c} J=\varphi \left(q\left(t_{c}+T_{o}\right)\right)+\int_{t_{c}}^{t_{c}+T_{o}}\left(L_{1}\left(\dot{q}\left(k\right)\right)+L_{2}\left(q\left(k\right)\right)+L_{3}\left(q\left(k\right)\right)\right)dk \end{array}$$
where $q={\left(\theta ,\dot{\theta }\right)}^{T}$$\in R^{2N_{j}}$ is the state vector of the manipulator, $\theta ={\left(\theta_{1},\theta_{2},\cdots ,\theta_{N_{j}}\right)}^{T}\in R^{N_{j}}$ is the vector composed of the joint angles of the manipulator, $N_{j}$ is the degrees of freedom of the manipulator, $T_{o}\left(T_{o}\leq T_{p}\right)$ is the length of the trajectory (in our experiments, we used $T_{o}=T_{p}$ as a rule of thumb) and $\varphi (q\left(t_{c}+T_{o}\right))$ is the terminal cost which prevents the calculated trajectory of the robot from diverging. It is expressed as:(3)$$\begin{array}{c} \varphi \left(q\left(t_{c}+T_{o}\right)\right)=\frac{1}{2}{\left({\mathrm{FK}}_{N_{j}}\left(q\left(t_{c}+T_{o}\right)\right)-x_{\mathrm{del}}\right)}^{T}R\left({\mathrm{FK}}_{N_{j}}\left(q\left(t_{c}+T_{o}\right)\right)-x_{\mathrm{del}}\right) \end{array}$$
where ${\mathrm{FK}}_{j}$ is the forward kinematics of the robot that transform the robot state q from joint coordinates to position $p_{j}$ and velocity $v_{j}$ in the workspace coordinates. *R* is the diagonal positive definite weighting matrix. $x_{\mathrm{del}}$ is the terminal state of the robot which is calculated based on the optimal delivery position and the predicted mean position of the worker. In this study, $x_{\mathrm{del}}$ becomes $x_{\mathrm{del}}={\left(\mu_{w}^{\left(t_{c}+T_{o}\right)}+p_{\mathrm{del}},0\right)}^{T}$, where $\mu_{w}^{\left(t_{c}+T_{o}\right)}$ is the worker’s predicted position at the end of the trajectory $\left(t_{c}+T_{o}\right)$, and $p_{\mathrm{del}}$ is the calculated delivery position for the worker’s observed position. We calculate $p_{\mathrm{del}}$ after each sampling interval and assume that the variation in $p_{\mathrm{del}}$ is negligibly small during the sampling interval of the sensing system, which is 30 ms.

$L_{1}\left(\dot{q}\left(k\right)\right)$, $L_{2}\left(q\left(k\right)\right)$ and $L_{3}\left(q\left(k\right)\right)$ are the stage costs which are expressed as: (4)$$\begin{array}{cc} \; & L_{1}\left(\dot{q}\left(k\right)\right)=\frac{1}{2}\sum_{j=1}^{N}r_{j}\left(B_{\mathrm{vel},j}\left({\dot{\theta }}_{j}\left(k\right)\right)+B_{\mathrm{acc},j}\left({\ddot{\theta }}_{j}\left(k\right)\right)\right) \end{array}$$(5)$$\begin{array}{cc} \; & L_{2}\left(q\left(k\right)\right)=w\sum_{j=1}^{N_{j}}\frac{1}{D_{M}\left({\mathrm{FK}}_{p,j}\left(q\left(k\right)\right),\mu_{w}^{\left(k\right)},\Sigma_{w}^{\left(k\right)}\right)}. \end{array}$$(6)$$\begin{array}{cc} \; & L_{3}\left(q\left(k\right)\right)=\frac{1}{2}\sum_{j=1}^{N}\left({\mathrm{FK}}_{p,j}\left(q\left(k\right)\right)-{\left(\mu_{w}^{\left(k\right)}+p_{\mathrm{del}}\right)}^{T}\right)\;Q\;\left({\mathrm{FK}}_{p,j}\left(q\left(k\right)\right)-\left(\mu_{w}^{\left(k\right)}+p_{\mathrm{del}}\right)\right) \end{array}$$

$L_{1}\left(\dot{q}\left(k\right)\right)$ is the stage cost to maintain the robot velocity and acceleration within their maximum limits. $B_{\mathrm{vel},j}\left({\dot{\theta }}_{j}\left(k\right)\right)$ and $B_{\mathrm{acc},j}\left({\ddot{\theta }}_{j}\left(k\right)\right)$ are defined as: (7)$$\begin{array}{c} B_{\mathrm{vel},j}\left({\dot{\theta }}_{j}\left(k\right)\right)=\left\{\begin{array}{c} \;\;\;\;\;\;\;\;\;0\;\;\;\;\;\;\;\;\;\;\;(||{\dot{\theta }}_{j}||\leq {\dot{\theta }}_{\max,j})\\ {\left(\left|\right|{\dot{\theta }}_{j}\left|\right|-{\dot{\theta }}_{\max,j}\right)}^{2}\;\;\;(||{\dot{\theta }}_{j}||>{\dot{\theta }}_{\max,j}) \end{array}\right. \end{array}$$
(8)$$\begin{array}{c} B_{\mathrm{acc},j}\left({\ddot{\theta }}_{j}\left(k\right)\right)=\left\{\begin{array}{c} \;\;\;\;\;\;\;\;\;0\;\;\;\;\;\;\;\;\;\;\;(||{\ddot{\theta }}_{j}||\leq {\ddot{\theta }}_{\max,j})\\ {\left(\left|\right|{\ddot{\theta }}_{j}\left|\right|-{\ddot{\theta }}_{\max,j}\right)}^{2}\;\;\;(||{\ddot{\theta }}_{j}||>{\ddot{\theta }}_{\max,j}) \end{array}\right. \end{array}$$
where ${\dot{\theta }}_{\max,j}$ and ${\ddot{\theta }}_{\max,j}$ are the maximum velocity and maximum acceleration of the *j*th joint, respectively.

$L_{2}\left(q\left(k\right)\right)$ is the stage cost that prevents the robot from hitting the worker. *w* is a weighting coefficient of this cost function. $D_{M}\left(x,\mu ,\Sigma \right)=\sqrt{{\left(x-\mu \right)}^{T}\Sigma^{-1}\left(x-\mu \right)}$ is the Mahalanobis distance that considers the variance of the probabilistic density distribution. Using the Mahalanobis distance between the predicted worker’s position distribution $N(\mu_{w}^{\left(k\right)},\Sigma_{w}^{\left(k\right)})$ and the end-effector position ${\mathrm{FK}}_{p,j}\left(q\left(k\right)\right)$ at step *k*, an artificial potential field is constituted according to the predicted variance. The artificial potential becomes wider in the direction of larger variance in the predicted position.

$L_{3}\left(q\right)$ is the stage cost to ensure that the robot follows the worker’s motion. *Q* is the diagonal positive definite weighting matrix. This cost function is responsible for the human-following motion of the robot based on the worker’s predicted trajectory as shown in Figure 3. For each time step of the predicted position distribution of the worker $N(\mu_{w}^{\left(k\right)},\Sigma_{w}^{\left(k\right)})$, the desirable state of the robot is calculated so that the robot’s endpoint follows the predicted mean position of the worker $\mu_{w}^{\left(k\right)}$ offset by the calculated delivery position $p_{\mathrm{del}}$.

Now we can define the optimization problem that will be solved by our proposed system.
$$\begin{array}{cccc} minimize & \; & & J\\ subject\;to & \; & & \dot{q}=f\left(q,u\right)\\ \; & \; & & q\left(t\right)=q_{\mathrm{cur}} \end{array}$$
where *f* denotes the nonlinear term of the robot’s dynamics, u is the input vector, and q(t) is the initial state of the trajectory which corresponds to the current state $q_{\mathrm{cur}}$ of the robot. To solve this optimization problem with the equality constraints described above, we use the calculus of variations. The discretized Euler–Lagrange equations that the optimal solution should satisfy are expressed as: (9)$$\begin{array}{ccc} & & q^{\left(k+1\right)}=q^{\left(k\right)}+f\left(q^{\left(k\right)},u^{\left(k\right)}\right)\Delta t_{s}, \end{array}$$(10)$$\begin{array}{ccc} & & q^{\left(t\right)}=q_{\mathrm{cur}}, \end{array}$$(11)$$\begin{array}{ccc} & & \lambda^{\left(k\right)}=\lambda^{\left(k+1\right)}-{\left(\frac{\partial H}{\partial q}\right)}^{T}\left(q^{\left(k+1\right)},u^{\left(k\right)},\lambda^{\left(k+1\right)}\right), \end{array}$$(12)$$\begin{array}{ccc} & & \lambda^{\left(t+T_{o}\right)}={\left(\frac{\partial \varphi }{\partial q}\right)}^{T}\left(q^{\left(t+T_{o}\right)}\right), \end{array}$$
(13)$$\begin{array}{c} \frac{\partial H}{\partial u}\left(q^{\left(k\right)},u^{\left(k\right)},\lambda^{\left(k\right)}\right)=0, \end{array}$$
where *H* is the Hamiltonian and is defined as: (14)$$\begin{array}{c} H\left(q,u,\lambda \right)=L_{1}\left(\dot{q}\left(k\right)\right)+L_{2}\left(q\left(k\right)\right)+L_{3}\left(q\left(k\right)\right)+\lambda^{T}f\left(q,u\right). \end{array}$$

The procedure for calculating the online trajectory is shown in Algorithm 3. After the sequential optimization based on the gradient decent, we obtain the optimal trajectory $\left(q^{\left(t\right)},q^{\left(t+1\right)},\cdots ,q^{\left(t+T_{o}\right)}\right)$. For detailed calculations, please refer to our previous study [13].
**Algorithm 3:** Robot Trajectory Generator**Input:** Target delivery position $p_{\mathrm{del}}$, Predicted worker’s trajectory $\left(N^{\left(t\right)},N^{\left(t+1\right)},\cdots N^{\left(t+T_{p}\right)}\right)$, Current state of the robot $q_{\mathrm{cur}}$, Max length of the robot trajectory $T_{o}$**Output:** Optimal trajectory is $\left(q^{\left(t\right)},q^{\left(t+1\right)},\cdots q^{\left(t+T_{o}\right)}\right)$1: Initialize the set of input vectors u2: $q^{\left(t\right)}⇐q_{\mathrm{cur}}$3: **while**$\;\Sigma_{k=t}^{t+T_{o}}\left|\frac{\partial H}{\partial u}\left(q^{\left(k+1\right)},u^{\left(k\right)},\lambda^{\left(k+1\right)}\right)\right|<\epsilon$
**do**4:  **while** $k=1\;to\;T_{o}$ **do**5:   $q^{\left(t+k\right)}⇐q^{\left(t+k-1\right)}+f\left(q^{\left(t+k-1\right)},u^{\left(t+k-1\right)}\right)\Delta t_{s}$6:  **end while**7:  **while** $k=T_{o}\;to\;1$ **do**8:   $\lambda^{\left(t+k-1\right)}⇐\lambda^{\left(t+k\right)}-{\left(\frac{\partial H}{\partial q}\right)}^{T}\left(q^{\left(t+k\right)},u^{\left(t+k\right)},\lambda^{\left(t+k+1\right)}\right)$9:  **end while**10:  **while** $i=1\;to\;T_{o}$ **do**11:   $s_{i}⇐{\left(\frac{\partial H}{\partial u}\right)}^{T}\left(q^{\left(t+i-1\right)},u^{\left(t+i-1\right)},\lambda^{\left(t+i\right)}\right)$12:  **end while**13:  u⇐u+cs14:  **end while**15:  **while**$k=1\;to\;T_{o}$
**do**16:   $q^{\left(t+k\right)}⇐q^{\left(t+k-1\right)}+f\left(q^{\left(t+k-1\right)},u^{\left(t+k-1\right)}\right)\Delta t_{s}$17:  **end while**

## 4. Experiment

### 4.1. Experimental Setup

To evaluate the performance of the proposed scheme in a real-world environment, we used the planar manipulator PaDY proposed in our previous study [7]. PaDY was designed to assist the workers of an automobile factory. A parts tray and a tool holder were attached to the end-effector of PaDY to store the parts and tools required for car assembly tasks. The robot delivers the parts and tools to the worker during the assembly process. For the details of the hardware design of PaDY, please refer to [7].

The proposed scheme was installed in a computer with an Intel Core i7-3740QM (Quad-core processor, 2.7 GHz) with 16GB memory. All calculations were done within 30 ms, the sampling interval of the sensing system that tracks the position of the human worker.

We designed an experiment to demonstrate the effectiveness of the worker’s motion prediction in the human-following behavior of our proposed scheme. Figure 4 shows the experimental workspace and Figure 5 shows the top view of the setup for this experiment. In this experiment, the worker needs to perform the following six tasks:1Tightening a bolt (Task 1);2Attaching three grommets (Task 2);3Attaching one grommet (Task 3, Task 4, Task 5, Task 6).

Each task is performed at a separate working position in the workspace. The experiment is carried out as shown in Figure 6. The experiment is carried out as follows.

1The experiment begins when the robot starts to approach the worker standing at the working position for Task 1. The worker takes a bolt and the bolt tightening tool from the robot (Figure 6a).2The worker performs Task 1 (Figure 6b).3The worker moves to the working position for Task 2 and the robot follows him. The worker returns the bolt tightening tool to the tool holder (Figure 6c) and picks up three grommets from the parts tray.4The worker performs Task 2 (Figure 6d).5The worker moves to the working position for Task 3 and picks up a grommet from the tray (Figure 6e).6The worker performs Task 3 (Figure 6f).7The worker moves to the working position for Task 4 and picks up a grommet from the tray (Figure 6g).8The worker performs Task 4 (Figure 6h).9The worker moves to the working position for Task 5 and picks up another grommet from the parts tray (Figure 6i).10The worker performs Task 5 (Figure 6j).11The worker moves to the working position Task 6 and picks up the last grommet from the parts tray (Figure 6k).12The worker performs Task 6 (Figure 6l) and this concludes the experiment.

We performed this experiment with four different participants (A, B, C and D) to evaluate the robustness of the system for different workers. Each participant is asked to perform the complete work cycle ten times. The first trial is performed without using the predicted motion of the worker. Whereas, in all other trials, the predicted motion of the worker is used and the worker model is sequentially updated after completing each trial.

For more details about the experiment, please see the Appendix A.

### 4.2. Tracking Performance

Figure 7a shows the estimated delivery position and the robot’s end-effector position for trial 1 of a participant when the robot’s motion is calculated based on the observed position of the worker without using the motion prediction. The black vertical lines show the time when the worker performs each assembly task. Figure 7b shows the estimated delivery position and the robot’s end-effector position for trial 10 of the same participant when the robot’s motion is calculated based on the predicted position of the worker using the proposed scheme.

At the beginning of the experiment, the robot is at its home position and the participant is at the working position for Task 1. The robot starts its human-following motion after arriving at the delivery position for Task 1 (at around 6 s in Figure 7a,b). We can see that the robot keeps following the participant during the whole experiment in both schemes (with and without the use of motion prediction).

The green line in Figure 7a,b shows the tracking error which is the difference between the delivery position and the end-effector position. We can see that the maximum tracking error is reduced from about 0.5 m to 0.3 m by using the motion prediction.

It is not possible to completely eliminate the tracking error since the manipulator used for the experiments has a mechanical torque limiter at each joint and the maximum angular acceleration without activating the torque limiter is 90 deg/s ${}^{2}$. In both Figure 7a,b, a large tracking error around 30 s can be observed. This is because the participant makes a large movement around 30 s and the robot cannot follow the participant because of its acceleration limit.

### 4.3. Cycle Time

Figure 8 shows the comparison of the cycle time of the four participants in each trial. We define cycle time as the time required for a participant to complete all six tasks of the assembly process. In Figure 8, the cycle time of each trial is normalized by the time of trial 1. Remember that motion prediction was not used in trial 1.

We can see that the cycle time for each participant decreases as the number of trials increases. The cycle time of trial 10 is reduced to 65.6–74.8% of the cycle time of trial 1. This shows that motion prediction can improve the performance of participants and help them complete the assembly process faster.

Note that the proposed system ignores the dynamics of the interaction between the robot and the worker, assuming that the worker is well trained and the behavioral dynamics of the worker with respect to the robot’s movements can be ignored. If the effects of the robot’s motion on the worker can be modeled, the system can better deal with the effect of the interaction between the worker and the robot and a further improvement in the worker’s time efficiency could be expected.

### 4.4. HRI-Based Cost

Table 1 shows the average and maximum HRI-based costs for each participant during the human-following motion of the robot. Since the HRI-based cost increases as the safety and comfort of the worker decreases, it is desirable to have a low HRI-based cost in human–robot collaboration.

In Table 1, we see that there are no significant differences in the average and maximum HRI-based costs between trial 1 (when motion prediction is not used) and trial 10 (when motion prediction is used) for all four participants. Therefore, we conclude that the proposed prediction-based human-following control reduces the work cycle time without adversely affecting the safety and comfort of the workers.

## 5. Conclusions

We proposed a human-following motion planning and control scheme for a collaborative robot which supplies the necessary parts and tools to a worker in an automobile assembly process. The human-following motion of the collaborative robot makes it possible to provide anytime assistance to the worker who is moving around in the workspace.

The proposed scheme calculates an optimal delivery position for the current position of the worker by performing non-convex optimization of an HRI-based cost function. Whenever the worker’s position changes, the new optimal delivery position is calculated. Based on the observed movement of the worker, the motion of the worker is predicted and the robot’s trajectory is updated in real-time using model predictive control to ensure a smooth transition between the previous and new trajectories.

The proposed scheme was applied to a planar collaborative robot called PaDY. Experiments were conducted in a real environment where a worker performed a car assembly process with the assistance of the robot. The results of the experiments confirmed that our proposed scheme provides better assistance to the worker, improves the work efficiency, and ensures the safety and comfort of the worker.

This scheme has been designed for a single worker operating within his/her workspace. It is not designed for the cases when multiple workers are operating in the same workspace, or when the worker moves beyond the workspace. Moreover, we did not consider the dynamics of interaction between the robot and the human, assuming that the human worker in the factory is well trained and his/her behavior dynamics to the robot motion is negligible. If the effects of the robot’s motion on the human can be modeled, the system can better deal with the effect of the interaction and further improvement in time efficiency could be expected.

We believe that the human-following approach has tremendous potential in the field of collaborative robotics. The ability to provide anytime assistance is a key feature of our proposed method, and we believe it will be very useful in many other collaborative robot applications.

## Figures and Tables

**Figure 1 sensors-21-08229-f001:**
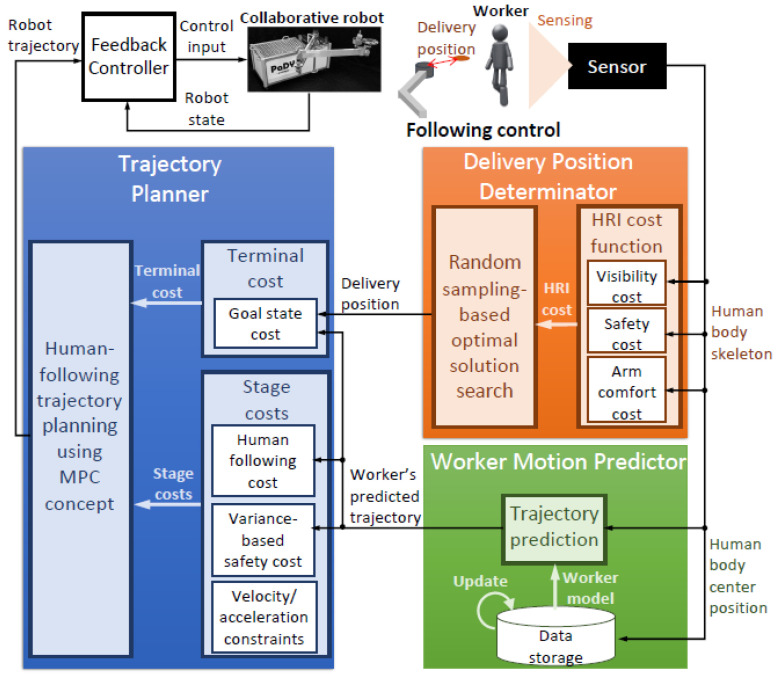
System architecture.

**Figure 2 sensors-21-08229-f002:**
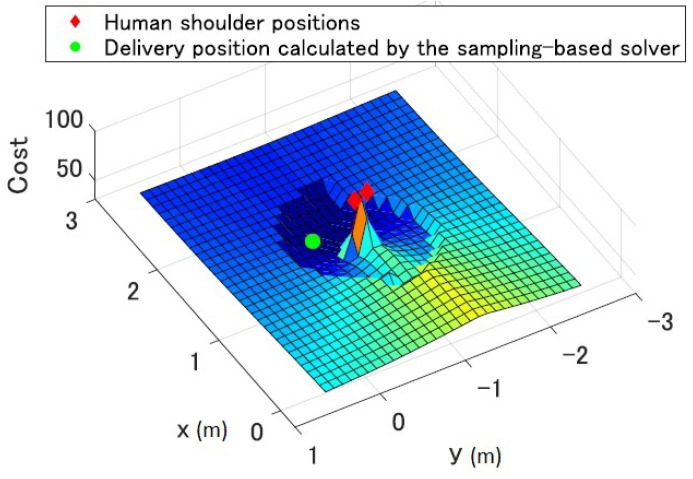
Example of the cost map calculated from the HRI constraints and its optimal delivery position.

**Figure 3 sensors-21-08229-f003:**
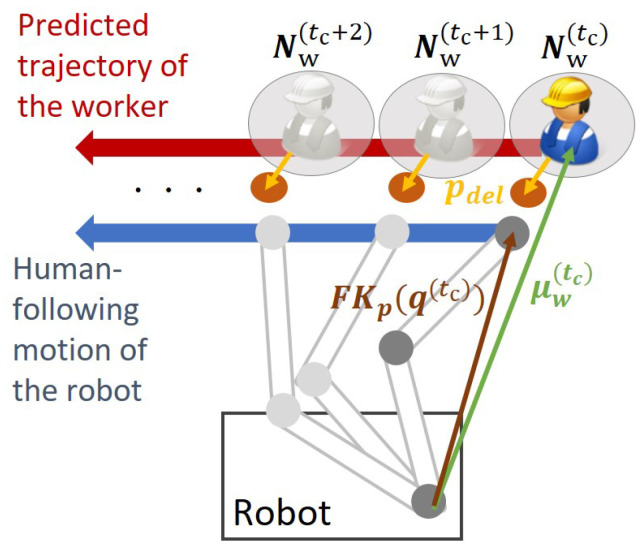
Concept of human-following motion planning based on the predicted trajectory of the worker.

**Figure 4 sensors-21-08229-f004:**
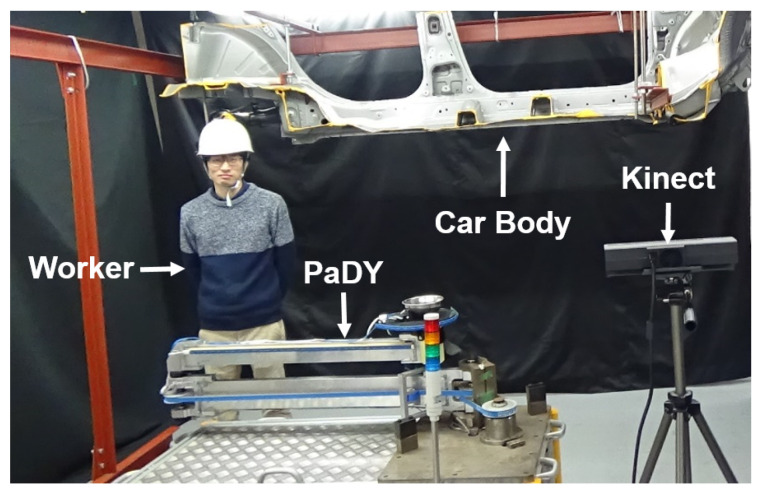
Experimental workspace.

**Figure 5 sensors-21-08229-f005:**
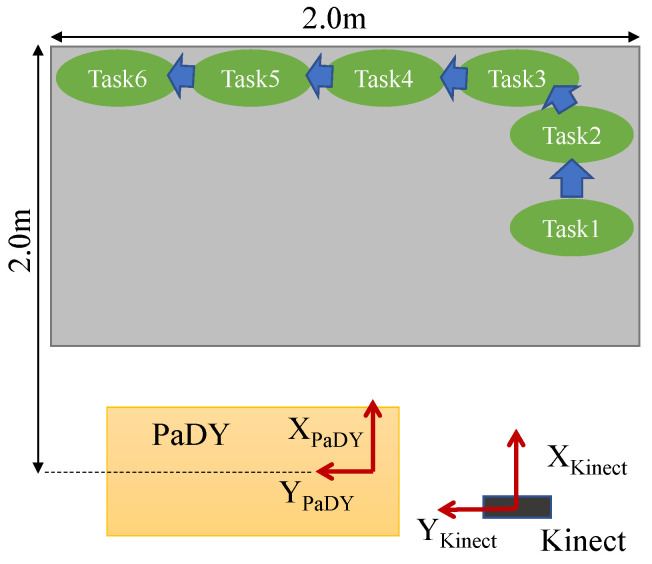
Top view of the experimental setup.

**Figure 6 sensors-21-08229-f006:**
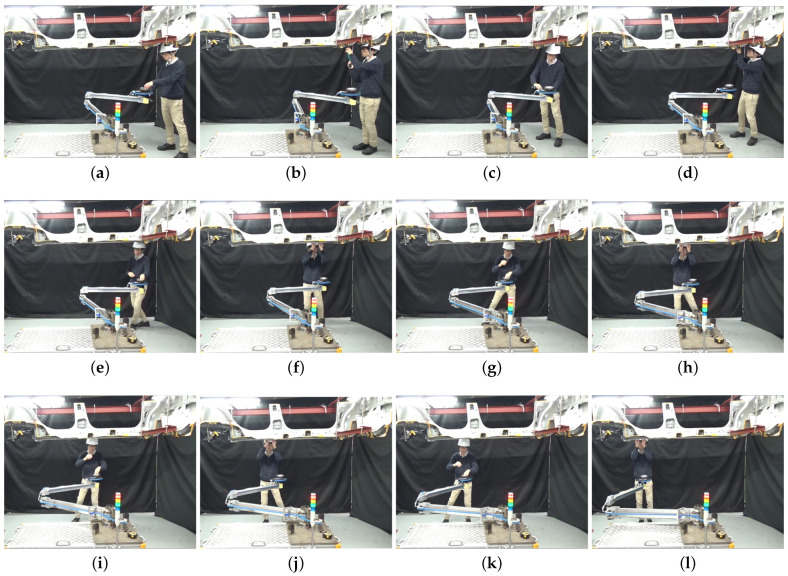
Experiment showing a complete work cycle where six tasks are performed. (**a**) A bolt and the tool are picked up; (**b**) Task 1 is performed; (**c**) The tool is returned and 3 grommets are picked up; (**d**) Task 2 is performed; (**e**) A grommet is picked up; (**f**) Task 3 is performed; (**g**) A grommet is picked up; (**h**) Task 4 is performed; (**i**) A grommet is picked up; (**j**) Task 5 is performed; (**k**) A grommet is picked up; (**l**) Task 6 is performed.

**Figure 7 sensors-21-08229-f007:**
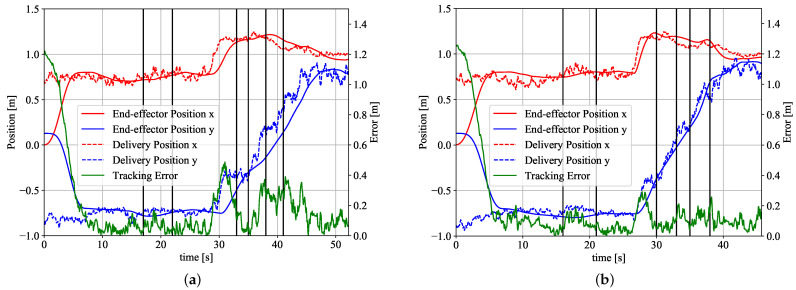
Tracking performance. (**a**) When motion prediction is not used; (**b**) When motion prediction is used.

**Figure 8 sensors-21-08229-f008:**
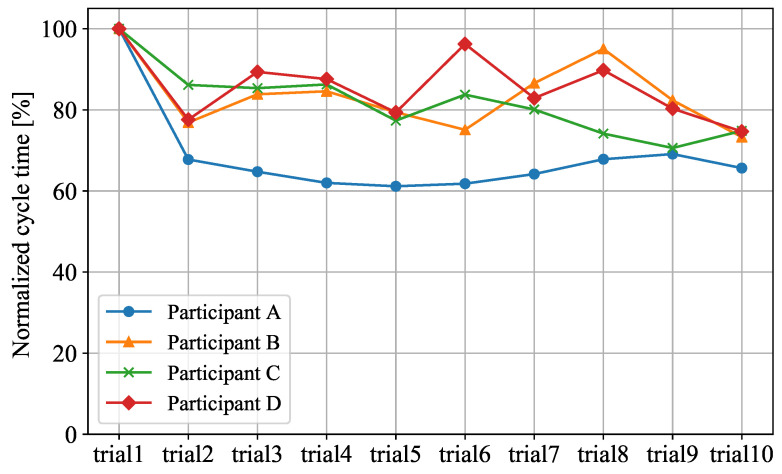
Comparison of Cycle Time.

**Table 1 sensors-21-08229-t001:** Summary of HRI-based costs during the human-following motion for each worker.

Worker	Average Cost (without Prediction)	Average Cost (with Prediction)	Max Cost (without Prediction)	Max Cost (with Prediction)
Worker A	8.99	11.79	36.34	35.82
Worker B	12.73	9.90	38.17	34.44
Worker C	18.35	13.56	39.65	31.33
Worker D	16.30	17.50	31.47	31.26

## Data Availability

The dataset generated from the experiments in this study can be found at https://github.com/kf-iqbal-29/Dataset-HumanFollowingCollaborativeRobot.git.

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
