# Peer review of "A Human-Following Motion Planning and Control Scheme for Collaborative Robots Based on Human Motion Prediction"

_sensors, 2021, doi:10.3390/s21248229_

Round 1

Reviewer 1 Report

Collaborative robots are a topic important in the HRI field and the industry. Authors used a model predictive control based trajectory planner to calculate the robot trajectory, which integrates a smart robot that can support the worker.

Suggestions:

Mention the type of methodology used and the scope. 

I recommend expanding and perhaps mentioning some of the limitations of the study.

Expand conclusions. There are other path planning algorithms, did the authors make a comparison of them with your proposal?

Author Response

Thank you very much for the review.

Here is a point by point answer to the reviewer’s comments:

Point 1:

Mention the type of methodology used and the scope. 

Response 1:

  • Methodology:
    We have added the following sentence in lines 69-70 to describe the methodology:

    “This is a quantitative study where we conducted the experiments ourselves and analyzed the data collected from these experiments to deduce the results.”

  • Scope:
    We have added the following sentences in lines 70-75 to describe the scope of our research:

    “This paper deals with the application of collaborative robots in industrial manufacturing. We have developed a safe and comfortable human-following motion planning and control scheme that delivers the required parts and tools to a factory worker in an automobile assembly process. The proposed scheme predicts the motion of the worker and calculates an optimal delivery position for the handover of part and tools from the worker to the robot for each task of the assembly process. This scheme has been designed for a single worker operating within his/her workspace. It is not designed for the cases when multiple workers are operating in the same workspace, or when the worker moves beyond the workspace.”

Point 2:

I recommend expanding and perhaps mentioning some of the limitations of the study.

Response 2:

We have expanded the conclusion section by adding the following sentences in lines 400-407 to explain the limitations of our scheme:

“This scheme has been designed for a single worker operating within his/her workspace. It is not designed for the cases when multiple workers are operating in the same workspace, or when the worker moves beyond the workspace.
Moreover, we have not considered the dynamics of interaction between the robot and the human, assuming that the human worker in the factory is well trained and his/her behavior dynamics to the robot motion is negligible. If the effects of the robot’s motion on the human can be modeled, the system can better deal with the effect of the interaction and further improvement in time efficiency could be expected.”

Point 3:
There are other path planning algorithms, did the authors make a comparison of them with your proposal?

Response 3:

As the reviewer said, there are many other path planning algorithms available. In this study, we have presented a review of the existing human-following motion planning schemes and a review of motion planning based on human motion prediction in section 2.2 and 2.3 respectively.

This study is a continuation of our previous research on collaborative robots. We have presented a detailed comparison with other motion planning schemes in our earlier studies such as doi: 1109/TRO.2019.2911800 and doi: 10.1504/IJMA.2021.113727

Reviewer 2 Report

The main goal is to algoritmically find the optimal position of the robot’s end effector by using the real-time worker data. At the same time, the future positions of the worker are predicted as probabilistic distributions. A Model Predictive Control (MPC) based trajectory planner is used to calculate a robot trajectory that supplies the required parts and tools to the worker and follows the predicted future positions of the worker.

The topic is really relevant to nowadays problems of robotics. The proposed appoach will improve the overal execution of the tasks of the cobots. Cobots are widely used nowadays in small to medium sized factories.

It adds model predictive control to the trajectory planning module of the cobot control module.

The methodology is fine. Conclusions consistent with the evidence and arguments presented. References are appropriate.

This paper can be accepted. Just check the URL at lines 396 and 397.

Author Response

Thank you very much for your review.
We really appreciate it and feel encouraged by it.

Here is a point by point answer to the reviewer's comments:

Point 1:
Just check the URL at lines 396 and 397.

Response 1:
As advised by the reviwer, we have corrected the URL at lines 396-397.

Reviewer 3 Report

Paper is very interesting and address state of art gap in Industry 4.0 & 5.0.

From my point of view, it is good for paper to put more light on some specific entity in the Industry ecosystem. Especially, on some specific categories of  people with special needs (blind and partially sighted, people in wheelchairs, etc.) in Industry ecosystem - Society 5.0 paradigma. Some paper  like (doi: ): 10.37528/FTTE/9788673954318/POSTEL.2020.020

Author Response

Thank you very much for your review.
Here is a point by point answer to the reviewer's comments:

Point 1:
From my point of view, it is good for paper to put more light on some specific entity in the Industry ecosystem. Especially, on some specific categories of  people with special needs (blind and partially sighted, people in wheelchairs, etc.) in Industry ecosystem - Society 5.0 paradigma. Some paper  like (doi: ): 10.37528/FTTE/9788673954318/POSTEL.2020.020

Response 1:
Our collaborative system has been designed for industrial manufacturing environments. We do not expect that the factory workers who will use our system will be blind, partially sighted or in wheelchairs.

We appreciate that the reviewer highlighted the importance of collaborative robots in Industry 5.0 environments. We have studied the paper referred by the reviewer and have added its citation in our paper with the following sentence in lines 30-31:
“Collaborative robots are expected to play a major role in the Industry 5.0 environments where people will work together with robots and smart machines [6].”